# Kinematic analysis of impairments and compensatory motor behavior during prosthetic grasping in below-elbow amputees

Amélie Touillet[1], Adrienne Gouzien[2], Marina Badin[1], Pierrick Herbe[1], Noël Martinet[1], Nathanaël Jarrassé[3], Agnès Roby-Brami[3]*

1 Louis Pierquin Centre of the Regional Institute of Rehabilitation, UGECAM Nord Est, Nancy, France,
2 Service de psychiatrie, Pôle Paris Centre, Hôpitaux de Saint-Maurice, Saint-Maurice, France, 3 Institute of Intelligent Systems and Robotics (ISIR), UMR 7222, CNRS/INSERM, U1150 Agathe-ISIR, Sorbonne University, Paris, France

* roby-brami@isir.upmc.fr

## Abstract

After a major upper limb amputation, the use of myoelectric prosthesis as assistive devices is possible. However, these prostheses remain quite difficult to control for grasping and manipulation of daily life objects. The aim of the present observational case study is to document the kinematics of grasping in a group of 10 below-elbow amputated patients fitted with a myoelectric prosthesis in order to describe and better understand their compensatory strategies. They performed a grasping to lift task toward 3 objects (a mug, a cylinder and a cone) placed at two distances within the reaching area in front of the patients. The kinematics of the trunk and upper-limb on the non-amputated and prosthetic sides were recorded with 3 electromagnetic Polhemus sensors placed on the hand, the forearm (or the corresponding site on the prosthesis) and the ipsilateral acromion. The 3D position of the elbow joint and the shoulder and elbow angles were calculated thanks to a preliminary calibration of the sensor position. We examined first the effect of side, distance and objects with non-parametric statistics. Prosthetic grasping was characterized by severe temporo-spatial impairments consistent with previous clinical or kinematic observations. The grasping phase was prolonged and the reaching and grasping components uncoupled. The 3D hand displacement was symmetrical in average, but with some differences according to the objects. Compensatory strategies involved the trunk and the proximal part of the upper-limb, as shown by a greater 3D displacement of the elbow for close target and a greater forward displacement of the acromion, particularly for far targets. The hand orientation at the time of grasping showed marked side differences with a more frontal azimuth, and a more "thumb-up" roll. The variation of hand orientation with the object on the prosthetic side, suggested that the lack of finger and wrist mobility imposed some adaptation of hand pose relative to the object. The detailed kinematic analysis allows more insight into the mechanisms of the compensatory strategies that could be due to both increased distal or proximal kinematic constraints. A better knowledge of those compensatory strategies is important for the prevention of musculoskeletal disorders and the development of innovative prosthetics.

**Data Availability Statement:** All relevant data are within the paper and its Supporting information files.

**Funding:** ARB and NJ received a funding by the Labex SMART (supported by French state funds managed by the ANR within the "Investissements d'Avenir" program under reference ANR-11-IDEX-0004-02). http://www.smart-labex.fr/ AG received an internship grant from the Institut Universitaire Ingéniérie de la Santé in Sorbonne University. https://iuis.sorbonne-universite.fr/ The funders had no role in study design, data collection and analysis, decision to publish, or preparation of the manuscript.

**Competing interests:** The authors have declared that no competing interests exist.

## Introduction

After a major upper-limb amputation, depending on the subject's expectations and life project, various prosthetic solutions are available with different control systems and terminal effectors. The patient's living conditions, leisure time and professional activity are considered to define the rehabilitation and fitting projects. The motor control of the prosthesis is not intuitive and requires learning. Body-powered prostheses are controlled through a harness connected by a cable that might provide for limited proprioceptive feedback [1]. Myoelectric control, which is the most common control mode, was invented in the fifties [2]. It associates the surface myo-electrical activities (EMG) from the residual limb to one or several prosthetic movements [3]. When several prosthetic joints have to be controlled, each of them is sequentially controlled by the same muscular contractions, with a switch between joint being activated by a co-contraction or specific contraction levels [4]. In addition, prosthetic users described that prosthetic use is complicated by the reduction of sensory information, particularly with current myoelectric prostheses which do not provide artificial sensory feedback [5, 6]. However, some limited perceptual information about the subject's environmental context and the manipulated object can be collected through the prosthesis by dynamic touch, that remains possible even in case of sensory impairment [7]. Visual control is often required during using prosthesis [8]. So, despite the potential possibilities offered by advanced prostheses such as polydigital hands, and despite the progress allowed by pattern-recognition techniques (allowing a more precise decoding of myoelectric signals and thus more controllable prosthetic joints [9]), their control remains particularly non-physiological (sequential and delayed) and complex both to learn and to use [10, 11]. The importance of the cognitive load necessary to a task also influences the strategies used [12].

All this leads to altered or unusual movements with other joints and segments when manipulating an object with a prosthetic device [13, 14]. These multiple limitations have a direct functional impact and are responsible for a high attentional load [6]. Moreover, this leads some amputees to abandon the use of a prosthesis [15] and particularly of upper-limb myoelectric prostheses [5].

Further studies are needed to better understand the difficulties of the amputees using prostheses in order to improve prosthetic solution. Most clinical methods used for the evaluation of prosthetic devices use specific outcome measures, psychometric scales [16] or standardized tests consisting in the manipulation of a panel of objects (e.g. SHAP Southampton Hand Assessment Procedure [17]). Precise methods are important for the clinical monitoring of the patients, in particular for the evaluation of new clinical or technological solutions, as well as for the future development of innovative prosthetic progress [18]. Instrumented laboratory methods with kinematic and/or kinetic recordings allow a better quantification of the task performance (review in [15]). Kinematic assessments using motion capture technologies have proven to be valuable for identifying movement strategies and also for assessing compensatory movements in upper limb amputees [13, 19, 20], including in patients with targeted reinnervation [21, 22].

Kinematic studies are now largely used in clinical research, in particular for the analysis of gait in lower-limb amputees [23] (see a review in [24]) but remain relatively scarce in the domain of upper-limb prosthetics. The reason is probably linked to the greater complexity of upper-limb actions (non-automatic, asymmetric and open chain) relative to lower-limb and to a lesser standardization of kinematic methods.

In addition, some kinematic studies in the domain of upper-limb prosthetics have been carried out with prosthetic simulators on able-bodied volunteers [1, 25–28].

The measurement of compensatory movements while moving objects with upper limb prostheses highlighted three categories of compensatory strategies following comparison of

transradial myoelectric prosthesis users with able-bodied subjects during bimanual tasks: pre-positioning of devices and objects in the workspace, posture compensations and a range of motion compensations [29].

The aim of the present study is to go further in the kinematic analysis of reaching and grasping by recording simultaneously the hand trajectory and joint rotations. Our perspective is to quantify both the alteration of movement quality during a simple goal-directed task and the amount of some proximal joints rotations in order to specify compensatory strategies. In particular we will study the impact on kinematic of the type of objects and of the distance between the subject and the object which have been mostly disregarded until now.

A better knowledge of compensatory strategies is important to understand how the motor system of amputees adapts to the loss of a part of a limb and its partial replacement with a prosthesis. From a clinical point of view, it is important to consider compensatory strategies and to differentiate between useful/unavoidable and harmful/avoidable ones. Indeed, certain compensations are inevitable when using prostheses. A better understanding of their mechanisms would help to manage compensatory strategies during rehabilitation for the prevention and treatment of musculoskeletal disorders. In addition, instrumented recordings may contribute to the evaluation of technical advances in prosthesis conception and control [30]. Additionally, a growing number of research projects in the development of advanced prosthetic control are being based on a better understanding of the compensatory movements and the pathological coordination strategies [31, 32].

## Methods

### Participants

A convenience sample of 10 participants with below-elbow amputation (9 males and 1 female aged from 22 to 61 years, mean 41.4 years, standard deviation 11.9) was included at the Regional Rehabilitation Institute (UGECAM Nord Est) in Nancy. The only inclusion criteria were amputation or congenital absence at the level of the forearm, in possession of a myoelectric prosthesis and normal, or corrected to normal, vision. Exclusion criteria were the presence of another neurological or orthopaedical pathology affecting the upper-limb. The protocol was approved by a local ethics committee (CER Paris Descartes) and all the subjects gave written informed consent prior to participating.

The main clinical data are indicated on Table 1. Nine right-handed participants were amputated after a traumatic injury, seven of their dominant right hand and two of their non-dominant left hand. The last patient suffered from right congenital upper-limb agenesis. The time elapsed since amputation as well as the cause and complexity of the prosthetic fitting varied greatly across participants (Table 1). The level of amputation also varied since four participants had a distal amputation level or radiocarpal disarticulation and six a middle or proximal forearm amputation. They were equipped with a myoelectric prothesis for 2.7 months to 27 years. The patients used their own prosthesis during the experiment. For all of them, prosthetic terminal effector was a tridigital prosthetic hand with one DOF (closing and opening tridigital pinch). Six of them, had a prosthesis with motorized wrist rotation, that they were free to use during the task. Only one participant with radiocarpal disarticulation (P8) keeps some physiological prono-supination.

In addition, the participants had a personal interview with AG, who is a psychiatrist, and responded to a questionnaire in order to evaluate the functional, aesthetic and psychological dimensions of the bodily integration of their prosthesis. The questionnaire and scale are fully described in [33]. Briefly, the scale included i) the time of wearing the prosthesis per day; ii) the evaluation of the compensation of the functional disability, based on the OPUS

**Table 1. Main clinical data.**

| ID | Sex | Age | Dominant side | Prosthetic side | Level | Delay (months) | Cause | Prosthetic use (months) | Wrist rotation |
|----|-----|-----|---------------|-----------------|-------|----------------|-------|-------------------------|----------------|
| P0 | M | 50 | Right | Right | Middle third | 23 | T | 21 | Mot. |
| P1 | M | 38 | Right | Right | Proximal third | 60 | T | 57 | Mot. |
| P2 | M | 44 | Right | Right | Proximal third | 172 | T | 170 | Mot. |
| **P3** | **M** | **43** | **Left** | **Right** | **Distal third** | **532** | **C** | **197** | **none** |
| **P4** | **F** | **22** | **Right** | **Left** | **Proximal third** | **31** | **T** | **26** | Mot. |
| P5 | M | 22 | Right | Right | Distal third | 2 | T | 1 | none |
| P6 | M | 48 | Right | Right | Proximal third | 128 | T | 120 | Mot. |
| P7 | M | 42 | Right | Right | Distal third | 34 | T | 31 | none |
| P8 | M | 44 | Right | Right | RCD | 45 | T | 42 | Physio. |
| **P9** | **M** | **61** | **Right** | **Left** | **Proximal third** | **130** | **T** | **125** | Mot. |

Bold: Non dominant amputated limb.

RCD: radiocarpal disarticulation, allowing physiological wrist rotation

T: traumatic, C: congenital.

Mot. Motorized wrist rotation

Physio: Physiological wrist rotation

questionnaire [34]; iii) subjective evaluations of the aesthetic and social discomfort related to several contexts; iv) the feeling of body integrity (the prosthesis is a constitutive part of themselves) and of indispensability (without it, they feel that something is missing). The items were weighted and normalized as described in [33] so that the maximum embodiment score was 10.

## Experimental set up and task

Participants were comfortably seated on a chair adjusted so that the table was approximately level with the navel, with the trunk free. The starting position was with the hand or prosthesis placed, with the fingers closed, on a mark on the table in the sagittal plane, the forearm was in mid-prone, the elbow flexed to ~90˚. (Fig 1). The tasks were carried out with eyes open.

The reference frame for the kinematic measures is indicated in red.

Reaching and grasping movements were evaluated with three different objects: a cylinder (height 0.15 m, diameter 0.04m, weight 0.3kg), a cardboard truncated cone (height 0.18 m, diameters 0.1m and 0.04m, weight 0.2kg) and a mug (height 0.10 m, diameters 0.9m, weight 0.32kg). These objects were chosen after discussion with occupational therapists examining the grasping affordances offered by various objects to anatomical or tridigital prosthetic hands.

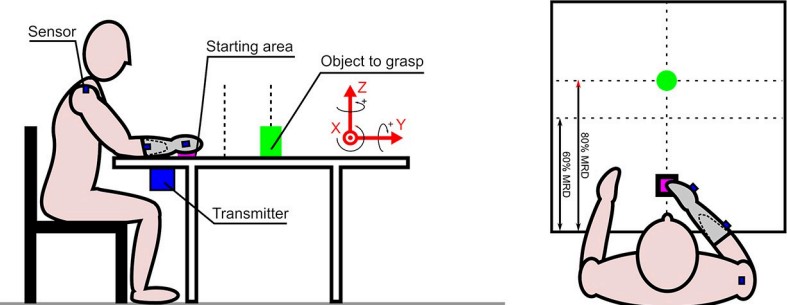

**Fig 1. Side view and horizontal view of the experimental set-up.**

The easiest object to grasp by amputees is a cone, routinely used in rehabilitation. In contrast, amputees have difficulties to grasp a mug by the handle, the cylinder being intermediate. The objects were placed on the table in the midline at two distances close and far, adjusted for each subject at respectively to 60% and 80% of the maximal reachable distance (MRD) of the prosthetic side. MRD was measured before the session when the participant was comfortably sitting, between the patient belly and the most distant forward distance he/she is able to reach with the centre of the prosthetic hand. The handle of the mug was oriented 45˚ by reference to the sagittal plane.

After the installation of the set-up, participants practiced several grasping movements to each object before recording. They were instructed to reach the object at a comfortable speed after the verbal signal of the experimenter, to grasp it and to lift above the table, then to put it back in the same position. No instructions were given regarding the way the objects should be grasped, allowing different hand orientation and height for grasping the cylinder and cone; the mug was oriented in a way to favour grasping by the handle [35].

The participants performed the experiment first with the non-amputated limb then with the prosthetic one. Then participants had to perform the task in the different Object-Distance conditions in a pseudo-random presentation order. Three repetitions were successively recorded for each condition. The data collection for each trial began at the verbal signal and lasted for 5 s. The sitting posture was regularly visually checked by the experimenter.

## Data collection

A 6-degree-of-freedom electromagnetic tracking device, the Polhemus Fastrak (SPACE FAS-TRAK, Colchester, VT, USA), was used to record the kinematic data at 30 Hz. This system gives position data and Euler angles (azimuth, elevation, roll) in a global coordinate system given by the Polhemus transmitter X rightward, Y forward, Z upward (The reported root mean square (RMS) accuracy of this system is 0.3–0.8mm for position and 0.15˚ for orientation when used within a 76-cm source to sensor separation, SPACE FASTRAK User's Manuel, Revision F. Colchester, VT; Polhemus Inc.; 1993). The transmitter was fixed under the table. One Polhemus sensor was attached with tape on the dorsum of the hand with its main axis along the third metacarpal bone (or similarly on the prosthesis), another to the dorsum of the forearm (or prosthesis). A third sensor was attached to the ipsilateral acromion.

These sensors directly give their position and orientation by reference to the reference frame of the transmitter: X laterally, Y forward and Z upward (Fig 1B). The 3D position of the elbow $P_{EL}$ was calculated thanks to a preliminary calibration procedure of the forearm segment (individual measurements of hand and forearm sensor localization along the forearm and elbow axis position). The humerus segment (vector $X_{arm}$) was then reconstructed as the segment linking the shoulder acromion ($P_{SH}$) and the centre of the elbow joint ($P_{EL}$). and the centre of the elbow joint. The 3D positions and orientations obtained for the left arm were mirrored to the right side for statistic comparisons.

For the calculation of joint angles, we chose the rigorous formalism of the ISB shoulder group [36]. The analysis was focused on shoulder elevation α and elbow extension β, which were computed the following way (Fig 2).

Shoulder elevation angle α was defined as the angle between the arm/humerus vector $\overrightarrow{X_{arm}}$ and the $\overrightarrow{P_H P_{SH}}$ vector representing the trunk. $\overrightarrow{X_{arm}}$ is the vector connecting the shoulder sensor (which centre is defined as $P_{SH}$) to the reconstructed elbow point $P_{EL}$ (which position is reconstructed in the forearm sensor frame located in $P_{FA}$ thanks to initial measurement of length lfa). $\overrightarrow{P_H P_{SH}}$ is the vector connecting the shoulder sensor centre $P_{SH}$ to the reconstructed centre

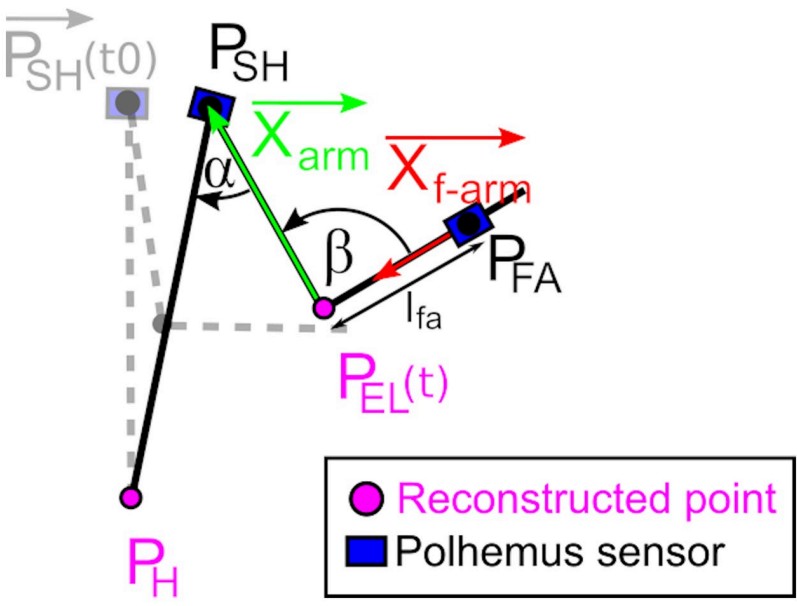

**Fig 2. Schematic representation -projected in sagittal plane- of the shoulder elevation angle α and elbow flexion-extension β.**

of the hip $P_H$ (with $\overrightarrow{||P_H P_{SH}||}$ norm being measured experimentally at the beginning of the experimental session, $P_H$ being the vertical projection of $P_{SH}(t0)$ on the horizontal plane to which the hip belong, and $P_H$ considered as fixed during the experiment). Thus,

$$\alpha(t) = \cos^{-1}\left(\overrightarrow{X_{arm}}(t) \cdot \overrightarrow{P_H P_{SH}}(t)\right)$$

Elbow extension angle β was defined as the angle between the previously defined arm/humerus vector $\overrightarrow{X_{arm}}$ and the vector $\overrightarrow{X_{f-arm}}$ which is one of the forearm sensor axis which was oriented specifically when placed on the participant limb so that it is aligned with forearm main axis. Thus:

$$\beta(t) = \cos^{-1}\left(\overrightarrow{X_{arm}}(t) \cdot \overrightarrow{X_{f-arm}}(t)\right)$$

The 3D positions and orientations obtained for the left arm were mirrored to the right side for ease of comparison.

## Data analysis

The aim of the present study is to describe the kinematic behaviour of a case series of amputated patients wearing a prosthesis. To that end, we quantified a series of kinematic dependent variables in order i) to compare the prosthetic and non-amputated sides, so that each participant is his own control and ii) to examine the interactions of the Factor Side (Prosthetic, non-amputated), with the factors Distance (Close-Far) and Object (3 levels).

## Dependent variables

The velocity profile of the hand (non-amputated or prosthetic) sensor was calculated by derivation of the displacement data and used to determine the timing of the tasks. The following

times were automatically determined, checked and eventually corrected visually thanks to an interactive home-made computer routine presenting simultaneously the movement trajectory and velocity profile (programmed in Labview©).

- The onset of movement (t0) was the first sample above a threshold of 0.05 m/s.

- The time (tv) of the maximum tangential velocity (Vmax) delimits the acceleration phase (between, t0 and tv).

- The time of grasping (tg) was chosen as a local minimum of velocity between the reach and lift velocity peaks coinciding with a reversal point of the hand trajectory. The reaching phase (between t0 and tg) includes both reaching and grasping unto the time of lifting.

In order to quantify the smoothness of reaching movements, the number of velocity peaks was calculated during the reaching to grasp phase using a velocity threshold of 0.05 m/s and a duration threshold of 100 ms (Labview©).

The spatial organization of movement was quantified by the 3D displacement of the hand/prosthesis. The curvature of hand trajectory was calculated as the ratio between the cumulated distance of the trajectory for reaching and the direct 3D distance (it is 1 if the trajectory is linear). The trunk and upper-limb involvement in the task were measured by the 3D displacement of the elbow and acromion and by the range of rotation in shoulder elevation and elbow flexion-extension during the reaching phase respectively.

The orientation of the hand for grasping was quantified by the Euler angles recorded at time tg: hand azimuth is the orientation of the hand in the horizontal plane (0˚ is oriented forward, 90˚ internally); pitch is the angle within the vertical plane defined by the azimuth (positive upward) and roll is around the longitudinal axis of the hand (0˚ thumb-up, 90˚ palmdown).

## Statistics

The independent factors were side (Prosthetic, non-amputated), distance (Close, Far) and Object (3 levels). Since the dependent variables were not normally distributed (Shapiro-Wills test, $p < 0.001$ for most variables) we used non-parametric statistical analyses.

The analysis was performed according to the following steps.

First, we analysed the effect of Side and Distance on the values obtained by averaging the dependent variables over the three objects. Friedman analysis was used to test the effect of condition (4 conditions: Side x Distance). When Friedman analysis showed significant variations, Wilcoxon test was used for paired comparisons: separately the effect of Side (for both distances) and the effect of Distance (for both sides).

Secondly: the effects of object were investigated by Friedman performed for each of the four combined side-distance conditions and completed by paired comparisons between objects (Wilcoxon test).

Effect sizes of paired comparisons were measured by the Cohen's d (difference between means divided by the standard error of the difference).

Correlation analysis was performed to test the relationship between clinical data (age, delay from amputation, duration of prosthetic use, embodiment score) and kinematic variables. Kinematic variables were expressed as a percent of variation of the prosthetic side (P) by reference to the non-amputated side (NA)

$$percent\ of\ variation = (P - NA) \div NA * 100$$

## Results

### Temporal organization of the hand movement. Effect of both side and distance

The kinematics of reaching and lifting are illustrated on Fig 3 in a representative participant amputated on the right side.

All the kinematic variables describing the temporo-spatial organization of the hand reaching movement varied significantly with the side and distance conditions (Friedmann 4 levels, $p<0.0001$), excepted the duration of the acceleration (Friedmann ns).

The maximum velocity of reaching was similar on both sides: there was no significant differences for the close target and borderline difference (Wilcoxon $p = 0.05$, $d = 0.7$) for the far target. The maximum velocity increased with distance both on the non-amputated (mean ± sem: $0.67 \pm 0.04$ m/s and $0.86 \pm 0.03$ m/s for the close and far targets, $d = 1.99$, Wilcoxon $p<0.005$) and the prosthetic sides ($0.66 \pm 0.04$ m/s and $0.79 \pm 0.04$ m/s, Wilcoxon $p<0.005$, $d = 1.62$) (Fig 4).

The box limits indicate the 25/75 percentiles and the whiskers the confidence interval. The median is indicated by the thick line. Dots indicate outliers.

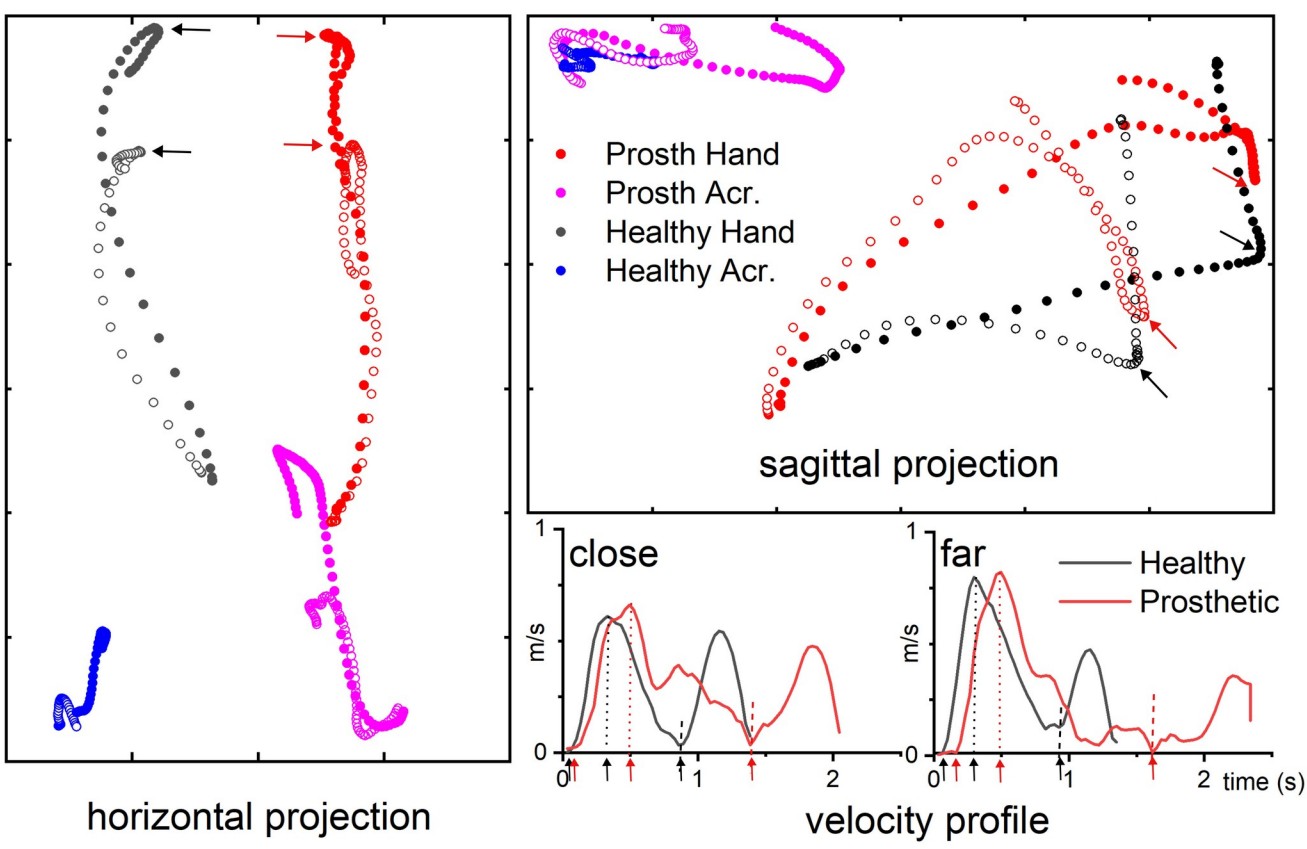

**Fig 3. Example of hand and acromion trajectories in a representative participant.** Horizontal (left) and sagittal (right) projections of the hand (black circles) and prosthesis (red circles) superimposed trajectories for the far (filled circles) and the close (open circles) targets, the object is a cylinder. The trajectories of the acromion sensor on the non-amputated (blue circles) and prosthetic (magenta circles) sides are also indicated. The corresponding velocity profiles are displayed on the right bottom corner for the close and the far targets. On the velocity profiles, black and red arrows indicate the times tr (onset of movements), tv (time of maximum velocity) and tg (time of grasping); the times tv and tg are also indicated by dotted and dashed lines respectively. The times tg are also indicated by arrows on the hand trajectories.

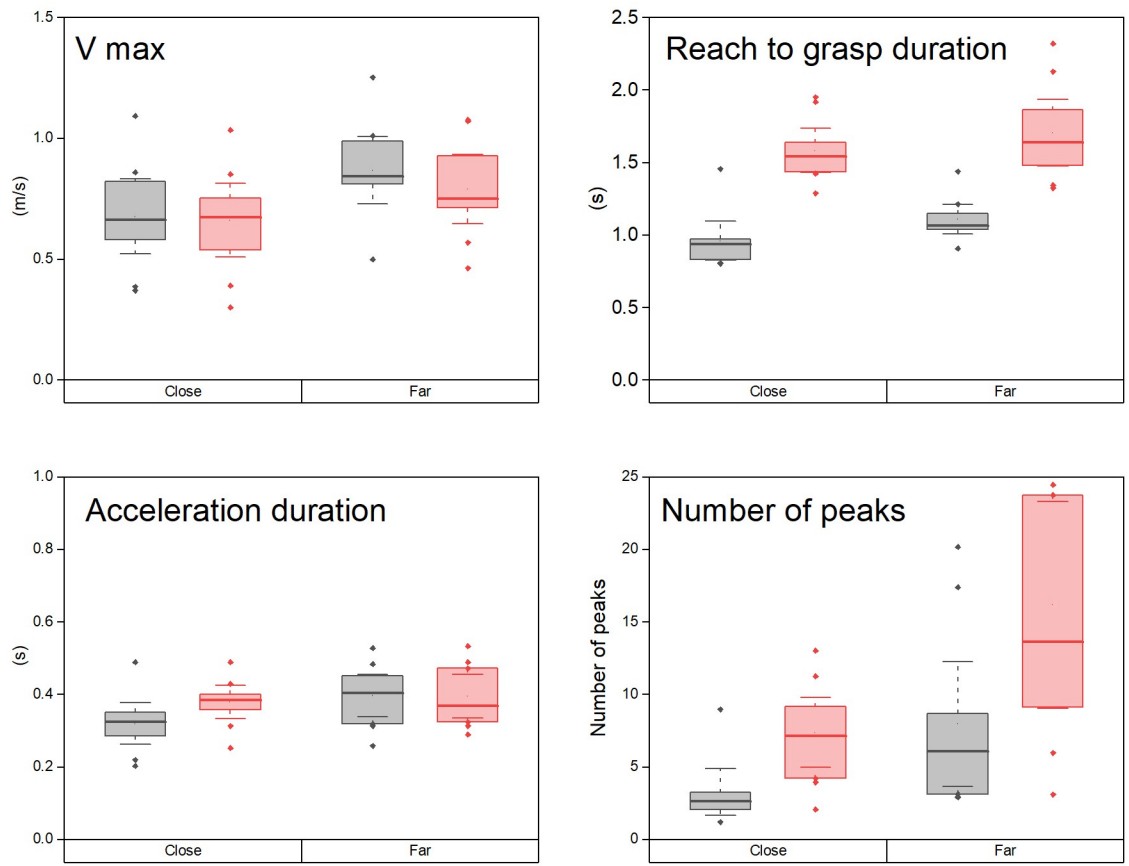

**Fig 4. Box plots comparing the temporal variables during reaching between the non-amputated (grey) and prosthetic (red) sides.**

The duration of the reaching phase was significantly shorter on the non-amputated side (0.96 ± 0.04s and 1.11 ± 0.03s), than on the prosthetic side (1.58 ± 0.04s and 1.70 ± 0.06s) (Wilcoxon p<0.005, d = 2.59 and d = 2.21 for the close and far targets respectively). On the non-amputated side, the duration of reaching was longer for the far than the close target (Wilcoxon p<0.007, d = 1.87). This scaling was not observed on the prosthetic side (Wilcoxon non significant).

The duration of the acceleration phase was not significantly different in the non-amputated and prosthetic sides. It was longer for the far than the close target on the non-amputated side (Wilcoxon p = 0.007, d = 1.19) but not on the prosthetic side.

The number of velocity peaks during the reach to grasp phase was significantly smaller on the non-amputated side (Wilcoxon, p<0.005, d = 195 and d = 1.52 for the close and far targets) but did not vary significantly with distance on either side.

## Shoulder and elbow range of rotation during reaching

The involvement of the upper-limb proximal joints was quantified by the range of rotation in the shoulder (alpha angle) and elbow (flexion-extension, FE). During reaching (Fig 5), both varied with the conditions (alpha: Friedman p = 0.002; elbow FE p = 0.00001).

The rotation in alpha angle increased with target distance without significant side difference. On the non-amputated side, it was 6.6 ± 2.2° and 19.5 ± 6° for the close and the far target

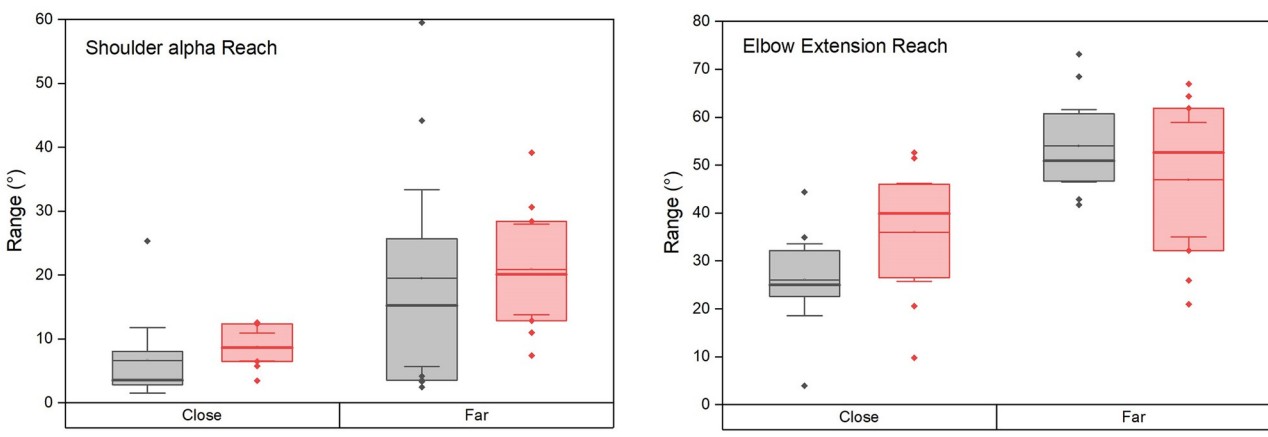

**Fig 5. Amount of shoulder and elbow rotations during reaching phases, same legend as Fig 4.** The thick horizontal line indicates the median and the thin one the mean.

(Wilcoxon p = 0.04, d = 0.76) and on the prosthetic side 8.7 ±1.0° and 20.9 ± 3.1° respectively (Wilcoxon p = 0.005, d = 1.33). The reaching movement was performed with elbow extension (positive values) which significantly increased with target distance. It was on the non-amputated side 26.0 ± 3.3° and 54.0 ± 3.3° for the close and the far targets (Wilcoxon p = 0.005, d = 2.97) and on the prosthetic side 36.0 ± 4.5° and 46.9 ± 5.3° respectively (Wilcoxon p = 0.005, d = 2.28). The effect of side was limited to a greater elbow extension on the prosthetic side for the close target (Wilcoxon p = 0.02, d = 1.18).

## Comparison of hand, elbow and acromion displacement during reaching

The displacement of the hand in the forward, upward and lateral directions varied with the side-distance condition (Friedmann p = 0.00002, p = 0.0003, p = 0.002 respectively, Fig 6, upper panels).

As expected, the mean forward displacement of the hand during reaching depended on target distance without side difference (on the non-amputated side it was 20.45 ± 2.3 cm and 33.2 ± 1.7, for the close and far target, Wilcoxon p = 0.005, d = 2.57, and on the prosthetic side 21.6 ± 1.9 cm and 33.4 ± 1.6 cm respectively, Wilcoxon p = 0.005, d = 2.66). During reaching, the hand raised more above the table for the far than the close target (on the non-amputated side it was 3.7 ± 0.5 cm and 7.6 ± 1.2 cm for the close and far target, Wilcoxon p = 0.005, d = 1.39 and on the prosthetic side 3.9 ± 0.7 cm and 9.8 ± 1.6 cm respectively, Wilcoxon p = 0.005, d = 1.59). The side difference was limited to a higher displacement on the prosthetic side for the far target (Wilcoxon p = 0.03, d = 0.76). The hand moved internally during reaching from its lateral initial position towards the midline (negative values). On the non-amputated side, this internal displacement was larger for the close than the far target (-7 ± 1 cm and -6.2 ± 0.9, Wilcoxon p = 0.02, d = 1.02) in contrast, there was no differences between target distances on the prosthetic side (-4.1 ± 1.3 cm and -4.9 ± 1.4 respectively). The side difference was limited to a smaller internal displacement for the close target on the prosthetic side (Wilcoxon p = 0.05, d = 0.83).

The curvature of the trajectory varied with the condition (Friedmann p = 0.0008). The trajectory in the horizontal plane was almost rectilinear (i.e. its value was ~1) for the far target on the non-amputated side (1.08 ± 0.02) and slightly curved for the close target (1.22 ± 0.08, p = 0.005). On the prosthetic side, the trajectory was curved for the far target (1.22 ± 0.04) and

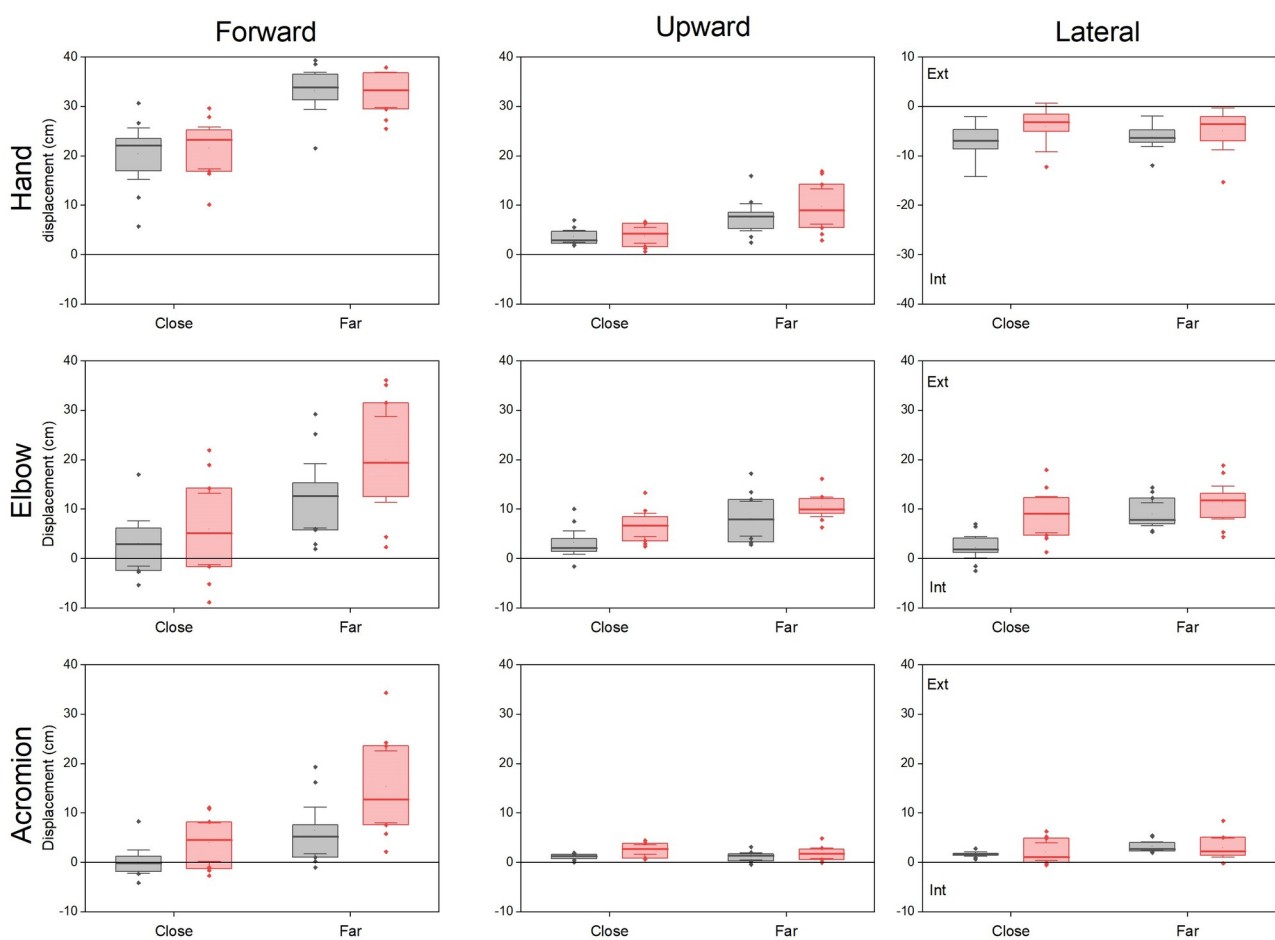

**Fig 6. Comparison of the hand, elbow and acromion displacements in the three directions during reaching.** Same legend as Fig 4.

even more for the close one (1.39 ± 0.08, p = 0.02). However, the greater curvature on the prosthetic side was only significant for the far distance (Wilcoxon, p = 0.02, d = 0.97).

The displacement of the elbow in the forward, upward and lateral directions during reaching varied with conditions (Friedmann p = 0.0002, p = 0.0005 and p = 0.0002 respectively, Fig 6, middle panels). The elbow displacement in those three directions was significantly greater for the far target for both sides (Wilcoxon p = 0.005). Significant side differences were observed for the close target: on the prosthetic side, the displacement of the elbow was higher (6.9 ± 1.0 cm versus 3.16 ± 1.1 cm, Wilcoxon p = 0.02) and more external than on the non-amputated side (8.9 ± 1.6 cm and 2.3 ± 1.0 cm respectively, Wilcoxon p = 0.01).

During reaching, the acromion moved mainly forward, with amounts which varied with side-distance conditions (Friedmann p<0.0002, Fig 6, lower panels). The forward displacement of the acromion increased with target distance on both sides (Wilcoxon, p = 0.005, d = 1.55 and d = 1.68 for the non-amputated and prosthetic sides and was larger on the prosthetic side for both targets. For the close target, the acromion remained stable on the non-amputated side (0.1 ± 1.0 cm) and moved forward (4.1 ± 1.7cm) on the prosthetic side (Wilcoxon p = 0.02, d = 0.98). For the far target, the forward displacement was smaller on the non-amputated (6.5 ± 2.1 cm) than on the prosthetic side (15.3 ± 3.2 cm, Wilcoxon, p = 0.005, d = 1.42).

## Effect of object

Visual observation showed that the mug was always grasped by the handle.

The effects of object were investigated separately for each side-distance condition by Friedman analysis (3 levels) completed by two by two comparisons (Wilcoxon test and Cohen's d).

The reach duration and maximum velocity of reaching varied with the object only on the non-amputated side for the close target (Friedman: p = 0.002 and p 0.007). In this later condition, the mug was reached with a significantly longer reach and slower velocity (0.61m/s ± 0.05) than the cylinder (0.70± 0.08 m/s, Wilcoxon p = 0.03, d = 0.73) and the cone (0.71 ± 0.08 m/s, Wilcoxon p = 0.005, d = 0.73).

The amount of hand displacement in the three directions during reach varied significantly with the objects in most side-distance conditions (Friedman ns to 0.0002). The statistical effects of object were similar in the non-amputated and prosthetic sides. The vertical displacement was higher for the cone (grand mean over conditions 8.1 ± 1.5 cm), than for the cylinder (6.3 ± 1.4 cm, d = 0.41 to d = 2.00) and the mug (4.3 ± 1.2 cm, d = 0.49 to d = 1.82). In most side/distance conditions the differences of forward and internal hand displacements were mainly observed between the mug and the two other objects (Wilcoxon p = ns to 0.005) the mug was grasped with a greater internal and shorter forward displacement than the cone or cylinder in all the side/distance conditions (Wilcoxon, p = 0.01 to p = 0.005, d = 0.88 to d = 2.23) there was no differences between the cone and cylinder.

## Hand orientation at the time of grasping, effect of object

When considering the mean of the three objects, azimuth and roll varied with the side and distance (Friedman: azimuth p = 0.0002, roll d = 0.0002) but not the pitch (Fig 7).

On the non-amputated side, the hand azimuth was directed roughly forward (0˚) for the far target (7.5 ± 3.0˚) and was increased (more frontal) for the close target (17.7 ± 3.7˚, Wilcoxon p = 0.005, d = 2.47). On the prosthetic side, the hand azimuth was also more frontal for the close relative to the far target (31.7 ± 3.9˚, Wilcoxon p = 0.005, d = 2.38). It was more frontal than on the non-amputated side for the far target (19.9 ± 3.5˚, Wilcoxon p = 0.005, d = 1.01) but there was no significant side difference for the close target. Concerning roll, on the non-amputated side the hand adopted an intermediate orientation between "thumb-up" (0˚) and "palm down" (90˚), which was slightly more rotated palm-down for the far target (32.7 ± 3.6˚ and 36.2 ± 3.6˚ for the close and the far targets, Wilcoxon p = 0.007, d = 1.69). On the prosthetic side, the roll was significantly less rotated than on the non-amputated side (9.5 ± 3.9˚ and 14.2 ± 2.9˚ for the close and far targets respectively; Wilkinson p = 0.007, d = 1.34 and Wilkinson p = 0.005, d = 1.70). On the prosthetic side, the roll did not vary significantly with target distance.

The hand orientation at the time of grasping depended on object shape differently according to the side. On the non-amputated side, it was quite independent of the object shape (Friedman analysis: significant difference only for pitch in the close target condition, p = 0.05). In contrast, on the prosthetic side there were significant differences for all the angles at both target distances: azimuth (Friedman p = 0.003 and p = 0.02 for close and far targets), pitch (Friedman p = 0.003 and p = 0.0001 respectively) and roll (Friedman p = 0.007 and p = 0.005 respectively). The mug was grasped more frontally than the cylinder (Wilkinson p = 0.05 and p = 0.04 for the close and far distances, d = 0.68 and d = 0.59) and the cone at close distance (Wilkinson p = 0.04, d = 0.77). The mug was grasped with a less "thumb-up" orientation than the cylinder (Wilkinson p = 0.02 and p = 0.05 for the close and far distances, d = 1.01 and d = 0.71) and the cone at close distance (Wilkinson p = 0.02, d = 1.22). On the prosthetic side, the pitch was larger for the cone and cylinder than for the mug for far targets (Wilcoxon

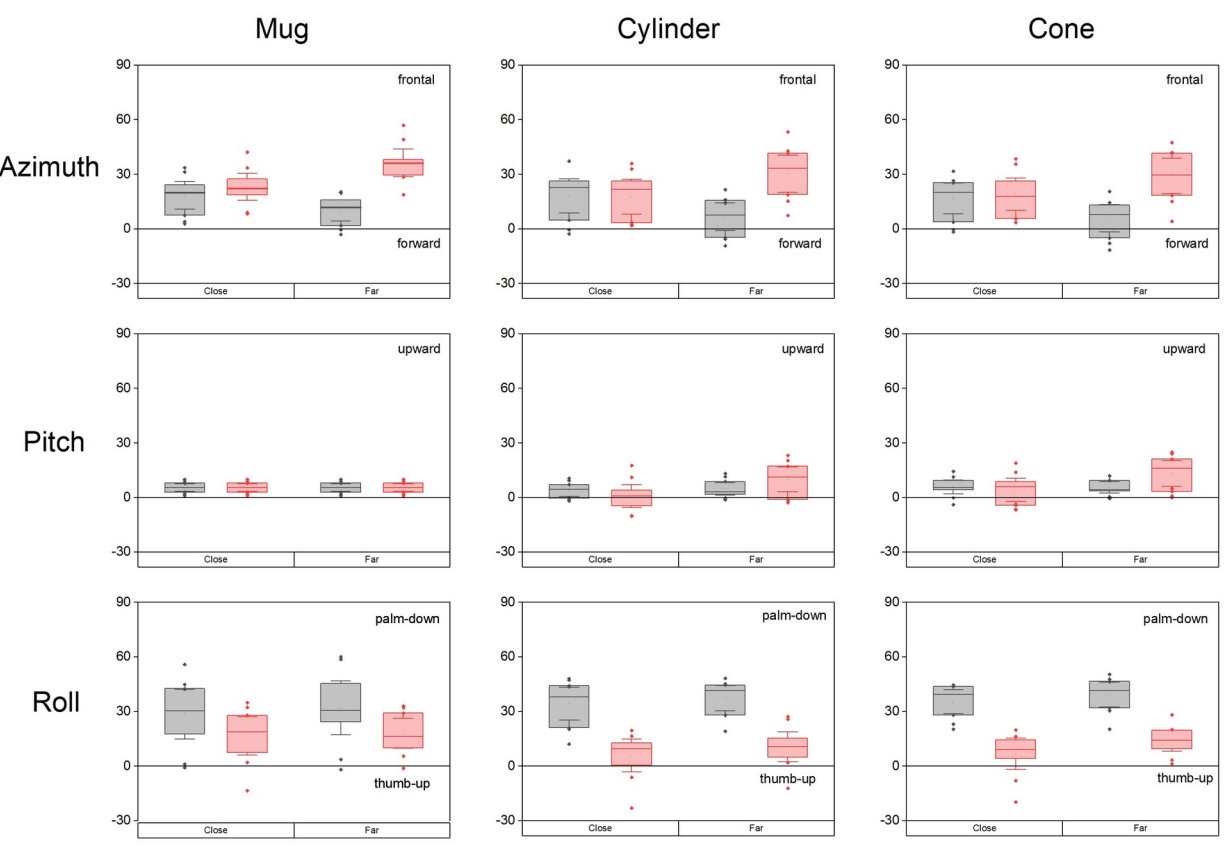

**Fig 7. Euler angles describing hand orientation at the time of grasping for the three objects.** Same legend as Fig 4.

p = 0.005, d = 2.03 and d = 1.76). On the non-amputated side, borderline differences were observed between the cone and cylinder (Wilcoxon p = 0.05, d = 0.66).

## Analysis of individual factors

Correlation analysis was performed between the embodiment score and the main kinematic variables. It showed that when the prosthesis was acceptably embodied, the duration of the reaching to grasp phase expressed as a percent of variation relative to the non-amputated side was shorter (p = 0.03) (Fig 8).

There was no significant relationship between the embodiment score and other kinematic variables. We explored potential relationships between kinematic variables, motor strategies (trunk flexion for reaching, particularities of hand orientation) and clinical data, in particular the proximo/distal level of the amputation and experience duration with a prosthesis but we did not find any pertinent relation with clinical variables. The limited number of participants did not allow to make further statistics.

## Discussion

To our knowledge, the present study is the first to analyse systematically the kinematics of grasping in order to compare the prosthetic and the non-amputated side with various objects placed at different distances. The kinematic analysis of hand trajectory during the reaching to grasp task demonstrates asymmetries between the non-amputated and prosthetic sides that

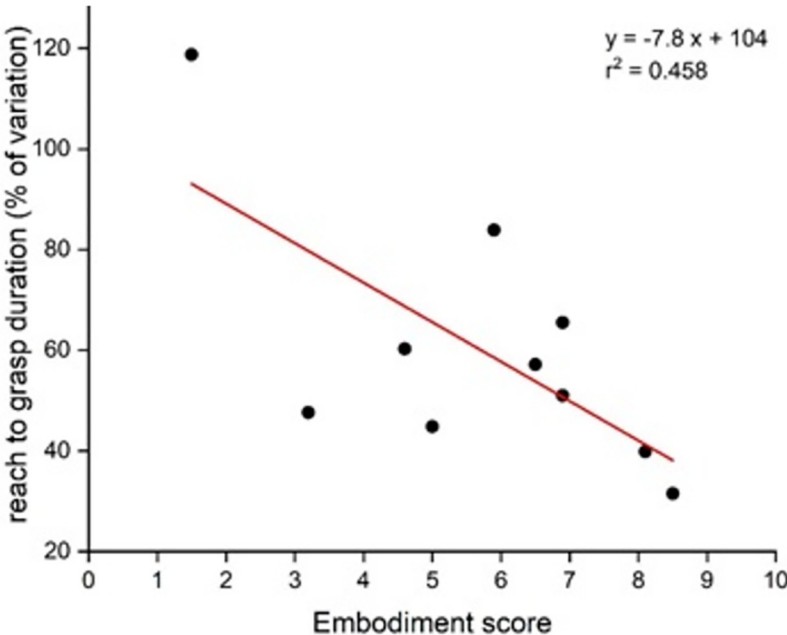

**Fig 8. Relationship between the embodiment score and the duration of the reaching phase, expressed as a % of variation relative to the non-amputated side.**

affect both the temporal and spatial organization of the gesture. In addition, the trunk and upper-limb coordination differed between sides, showing that the use of a prosthesis impacts the proximal limb and the global body coordination. The proximal changes could be both direct consequences of the distal constraints brought by the prosthesis and/or motor strategies aiming at compensating these consequences. The extensive literature on the kinematics of reaching to grasp in healthy subjects contrasts with scarce observations in amputees using a prosthesis. We shall examine successively the temporal and spatial organization of the movement. Temporal organization.

The temporal organization of motion relies on the trajectory of the end-point of the limb during goal-directed movements. Physiological goal directed upper-limb movements are characterized by well-known characteristics. The trajectory of the end-point is roughly linear during pointing with a smooth bell-shaped velocity profile scaled to movement distance [37]. These features evidence some optimal anticipation of biomechanical constraints [38]. Prehension was particularly described in Jeannerod's pioneering works on precision grip [39, 40], review in [41]. He distinguished the two components of prehension, reaching and grasping. Reaching is characterized by a smooth but asymmetric velocity profile, scaled to target distance. Preparation to grasp is performed in parallel with the fingers opening during reaching (preshaping) then closing at the termination of reach. Preshaping of fingers aperture is also observed during whole hand grasping [42]. The tight temporal coupling between reaching and grasping was confirmed by many experiments with perturbations [43–47]. Kinematic studies in amputees showed that they were able to perform smooth horizontal pointing task, for example with a robotic manipulandum (InMotion2 Shoulder-Elbow Robot®) [48]. In contrast, prosthesis users perform movements with a degraded quality when it comes to grasp and manipulate objects.

The present study confirms and specifies previous observations since the delay before lifting was massively increased on the prosthetic side. In contrast, the initial reaching component was

little affected: the duration of the acceleration phase and maximum velocity were similar on both sides. The maximum velocity (but not the acceleration duration) was scaled to target distance on both sides. This suggests that the reaching and grasping components were not performed in parallel but sequentially, consistently with [49–51]. Certainly some authors describe that non-sequential prosthetic movement is possible with Wing and Fraser observing a child (13 years old) with a congenital absence of a fore-arm who used a mechanical prothesis since her early age; she could pre-shape her prosthesis during a slower reaching followed by a faster closure than on the non-amputated side [52, 53]. However, more recent studies in adults showed that the prosthetic reaching movements were generally slower, less rectilinear and less smooth than physiological movements [49–51] with a temporal decoupling of the reaching and grasping components [49, 51]. Sequential control of prosthetic hand opening and pronosupination could participate in this phenomenon.

The increased number of velocity peaks on the prosthetic side suggests that grasping was prolonged with iterative small hand displacements related to difficulties for moulding the prosthesis around the object because of non-adaptative grip and smaller hand opening. The mechanical structure of the prosthetics (and its unbalanced weight repartition), obvious difficulties to operate the artificial control and impairment of direct somatosensory and proprioceptive feedback during interactions with the object, imposing permanent visual monitoring probably contribute to the slowing and irregularities of the grasping [54]. Moreover, the loss of mobility at wrist and finger levels probably participates to the impairment of grasping [28]. Kontson evaluated grasping simulating the limitations induced by conventional prostheses with bracing and strapping of able-bodied subjects performing Box and Block test. The simulation showed a decrease in performance [27, 55].

There are few kinematic studies of prosthetic reaching and grasping and most include a very small number of heterogeneous participants. The study by Martinet et al. [51] is a single case. Bouwsema et al. [49] included six participants three of them had upper arm amputation and used a combination of myoelectric hand and mechanical elbow (hybrid control), the three others had a transradial amputation and used myoelectric prosthesis. Engdahl et al. [50] included nine participants with transradial amputation using myoelectric or body-powered prostheses. So that, the mechanism of the decoupling between reaching and grasping remains unclear. The quality of prosthetic prehension movements depends on training as shown by studies in healthy subjects learning to use a prosthesis simulator with myoelectric [25] or body-powered control [1, 56]. In all cases, training reach to grasp tasks resulted in faster movements and shorter grasping delay.

Interestingly, the duration of the reach to grasp phase was correlated with a score evaluating the quality of embodiment [33] (see [57] for a similar evaluation of embodiment). However, the direction of the causal relationship is equivocal. Grasping and manipulation is at the basis of daily life activity. Is the feeling of embodiment determined by the duration of increased delays in performing daily activities? Or a lack of embodiment contributes to difficulties in motor control resulting in slower actions?

## Spatial organization of hand trajectory and inter-joint configuration for grasping

The spatial trajectory of the hand during reaching to grasp has been largely documented in non-amputated subjects but has been little described before in participants wearing a prosthesis [13, 30]. In the present study, the trajectories of the prosthetic and non-amputated hands were globally similar, in particular the forward displacement of the hand was identical on both sides. However, on the prosthetic side the trajectory was more curved according to object

distance as already observed [50]. The global hand displacement during reaching was slightly higher and less internally on the prosthetic side.

As largely observed in healthy subjects, the trajectory of the hand is due to some refined inter-joint coordination [58] involving most of the DoF of the upper-limb [59]. Coordination between the trunk, shoulder and elbow are particularly remarkable [60].

The use a prosthesis impedes or limits hand movement, wrist movement and prono-supination. Elbow movements can also be limited obviously if the amputation is transhumeral or for below elbow amputees because of the mechanical constraints of the socket. The present study allows to quantify the greater involvement of proximal limb and global body, confirming previous clinical and kinematic studies [13, 19, 30, 61, 62]. Amputees recruit additional DoF and/or increase the range of motion in the proximal joints and trunk (shoulder, torso) by reference to healthy subjects to compensate for the distal motor limitations [13, 61]. Those movements are specific and appropriate to the task, demonstrating that they are compensatory in nature as widely reported in clinical studies [63]. Montagnani evaluated compensation during grasping in able bodied subjects simulating the limitations induced by conventional prostheses with several orthosis. The compensations are more important with the orthesis which limits an higher number of DoF. He described that the presence of wrist flexion within the orthesis improves gesture efficiency and limits compensations [28].

The present study confirms that the involvement of the trunk and upper-limb joint rotations were quite different between sides. On the non-amputated side, the reaching movement was performed with a shoulder rotation ($\alpha$ angle bringing the arm away from the body) and an elbow extension $\beta$ both scaled with target distance; the trunk contributed to hand movement only for far target. The trunk and upper-limb coordination were modified by the use of a prosthesis. For close targets, the movement was performed with a greater range of shoulder and elbow rotations (significant only for elbow extension). The shoulder range of motion was particularly variable, consistently with previous studies [13, 19, 30]. The increased contribution of shoulder joint is also supported by the greater upward and external displacements of the elbow on the prosthetic side. In addition, the far target was reached with a supplementary forward displacement of the acromion corresponding to an increased flexion and/or axial rotation of the trunk (including scapula) as previously clinically described [63] and quantified by kinematic analyses during goal directed activities [13, 19, 62].

The sitting position used in the present study is probably more constraining than the standing position. Indeed, in standing position, trunk inclination was favoured [27] while in sitting position, compensations via movable body parts (at the trunk-head-upper limb level, including shoulder abduction) are even more necessary.

The compensatory strategies impose increased workload on proximal joints, including trunk and neck, and on the non-amputated side. The repeated use of postures with trunk torsion, anteflexion and lateral inclination, and with abduction-internal rotation of the shoulder are known as risk factors for musculoskeletal disorders [64] which can contribute to the handicap [65–69]. Indeed it is demonstrated that significant shoulder abduction is a source of major shoulder strain that can, in the long-term, cause pain [70] Similarly, prolonged arm elevation may result in shoulder pain and musculoskeletal disorders at various thresholds [71, 72].

## Hand orientation for grasping to lift

The present study brings additional information on motor behaviour by quantifying the hand orientation at the time when the object is grasped stably so that the participant can possibly initiate a lifting action. The hand orientation for grasping was quite asymmetric. On the non-amputated side, it varied little with the condition: the hand was slightly elevated above the

horizontal with an intermediate roll and its azimuth was scaled to target distance, consistently with [73].

On the prosthetic side, the azimuth at the time of reaching was more frontal and particularly sensitive to distance. The pitch was greater, particularly for the far target.

In addition, the orientation of the prosthetic hand at the time of grasping, as well as the upward and lateral displacement of the hand during reaching, were sensitive to the shape of the object. A stereotyped "thumb-up" orientation of the prosthesis (with roll tending to 90˚) was preferred for the cone and the cylinder, the cone was grasped at the highest height with the greatest pitch. Despite the fact the mug could be grasped in a variety of ways [35] it was always grasped by the handle, probably due to implicit understanding of the task. The specific shape of the mug's handle probably constrained the pose of the hand, which was grasped at the lowest height, with the most frontal azimuth, intermediate roll and without pitch. This suggest that on the non-amputated side, the different object shapes are taken over by the versatility of the fingers [74], so that they grasp objects with the same position and orientation of the hand relative to object position. In contrast, on the prosthetic side, the participants have to adapt their hand pose for grasping as a function of object shape.

## Mechanisms of compensatory behavior

The influence of the distal impairment on the proximal coordination is quite complex and two mechanisms can be hypothesized.

On one side, the shape of the object relative to the configuration of the prosthesis itself, could influence the choice of a given orientation of the extremity (e.g. more frontal azimuth and/or "thumb-up" posture). Prosthetic hand opening is smaller than in a healthy hand and in our study, only one kind of grip is possible: tridigital. The lack of finger and wrist mobility and of prono-supination for prosthesis users impose increased constraints on the upper-limb [13, 19, 62]. Then the choice of a functional hand pose would determine the trajectory of the limb for reaching and the trunk and upper-limb rotations in order to remain in a comfortable configuration for lifting [13, 19, 62] Indeed, the choice of a more frontal hand orientation imposes an elbow flexion and thus some trunk flexion to preserve the reaching distance since the prosthesis has no wrist mobility.

On the other side, the amputation may directly affect the proximal shoulder/elbow coordination. The added distal weight of the prosthesis may incite the participants to decrease the length of the lever arm by bending the trunk in order to decrease the torque constraints on the shoulder. Indeed, the weight distribution of the prosthesis differs from that of the missing limb with a shift in weight towards the most distal part of the prosthesis. This is due to the prosthetic hand and the motor which are the heaviest parts of the prosthesis and are the most distal. This difference may contribute to the asymmetry observed between amputated and intact limb kinematics.

The possible limitation of elbow extension might be due to the structure of the socket that may encompass the elbow to maintain the prosthesis particularly when the amputation of the forearm is proximal. Some conflicts may also impede the flexion of the elbow to reach the close target. In addition, acquired musculo-skeletal disorders may complicate the clinical picture [67, 69]. The two most frequent musculo-skeletal disorders observed in case of unilateral upper limb amputation are contralateral carpal tunnel syndrome and homolateral shoulder pain, which affect about 40% of amputees [65]. Whatever its initial mechanism, the lessening of elbow extension compensated by the increased recruitment of trunk flexion can be compared to the motor condition in stroke patients [75–78]. In both cases, trunk flexion is likely an exaggeration of the physiological coordination used to reach objects close to the maximum arm length [60].

Further studies are needed to determine, in each patient, the mechanisms of compensatory strategies since distal and proximal constraints may be combined [60]. Whatever their distal or proximal origin, kinematic modifications are probably learned skills, as suggested by Carey [19] who observed that compensations were greater in amputated prothetized participants than in non-amputated subjects braced. Accordingly, Major described that prosthesis experience had a strong positive relationship with average kinematic repeatability [13].

## Limitations

Our preliminary observational study includes a small convenience sample of participants which does not allow to stratify the population in homogeneous sub-groups. It would have been interesting to measure the maximal extension and flexion of the elbow with the prosthesis and the posture of the elbow joint when measuring maximal reaching distance to quantify the limitation of elbow mobility by the socket.

The participants were their own controls. The recruitment of a control sample of non-amputated subjects (matched in age, gender and laterality) would increase the power of our study.

Moreover Metzger [48] found that reaching task in vision condition are modified for both arms of amputee subjects compared to healthy subjects.

The biomechanical methods were limited. The opening-closing of the non-amputated or prosthesis hand and the use or not of motorized pronosupination were not measured. The unique sensor placed at the level of the acromion does not allow to differentiate the movements of the trunk from those of the shoulder complex and external/internal rotation of the upper-arm could not be calculated. It would be interesting to add an axial sensor at the level of the spine in order to differentiate the compensations performed at the level of trunk (torsion and inclination) from those performed at the level of shoulder complex (elevation and protraction of the scapula) and to improve the biomechanical model in order to measure glenohumeral rotation.

## Perspectives

Precise kinematic analyses of the compensatory movements are important for the prevention and therapy of musculoskeletal disorders which are particularly frequent in amputated patients. The repeated use of postures with trunk torsion, anteflexion and lateral inclination of the trunk, and with abduction-internal rotation of the shoulder are known as risk factors for musculoskeletal disorders [79]. So, a better knowledge of these parameters is necessary.

A growing number of studies are now evaluating the quality and the performance of the motor assistance provided by technical devices such as prostheses through the quantification of compensatory strategies [80]. The amount of compensatory strategies can also be used to drive the adaptation of a control mode and personalize the behaviour of the prosthesis to the user [81]. More recently, some researches started to take even more advantage of those motor compensations by closing the prosthesis control loop on them (i.e. considering them as an «error»), to cancel them (by reconfiguring the kinematics of the prosthesis) while allowing user to perform a task in a more ergonomic way [31]. Those researches highlight the necessity of a better knowledge and understanding of patient's particular kinematic strategies.

It would be important to have simplified recording devices usable in clinical routine to evaluate the compensations used by prosthetized patients in work or leisure situations. Further development are needed with inertial sensors in order to record longer activities in more ecological situations. Similarly, this method could be useful to evaluate other innovative medical devices such advanced polydigital hand [82], more flexible wrist prostheses, or the impact of

artificial proprioceptive feedback. Kinematic analysis could be operational for the choice and prescription of prosthesis and in order to guide rehabilitation techniques while preventing musculoskeletal disorders. Indeed, the rehabilitation program focusing on compensatory movements of prosthetic users should be personalized. A better understanding of compensatory strategies that would help discriminate between useful/unavoidable and harmful/avoidable ones, would allow more efficient rehabilitation care in upper limb amputees and improve their autonomy with respect to prosthetic settings.

## Supporting information

**S1 Data.**
(XLSX)

## Acknowledgments

The authors thank all the participants to this study.

## Author Contributions

**Conceptualization:** Amélie Touillet, Adrienne Gouzien, Noël Martinet, Nathanaël Jarrassé, Agnès Roby-Brami.

**Data curation:** Adrienne Gouzien, Marina Badin, Agnès Roby-Brami.

**Formal analysis:** Nathanaël Jarrassé.

**Funding acquisition:** Adrienne Gouzien, Nathanaël Jarrassé.

**Investigation:** Amélie Touillet, Adrienne Gouzien, Marina Badin, Noël Martinet, Nathanaël Jarrassé.

**Methodology:** Pierrick Herbe, Agnès Roby-Brami.

**Project administration:** Agnès Roby-Brami.

**Resources:** Amélie Touillet, Pierrick Herbe, Noël Martinet.

**Software:** Nathanaël Jarrassé, Agnès Roby-Brami.

**Supervision:** Amélie Touillet, Noël Martinet, Nathanaël Jarrassé.

**Validation:** Amélie Touillet, Nathanaël Jarrassé, Agnès Roby-Brami.

**Visualization:** Nathanaël Jarrassé, Agnès Roby-Brami.

**Writing – original draft:** Agnès Roby-Brami.

**Writing – review & editing:** Amélie Touillet, Adrienne Gouzien, Noël Martinet, Nathanaël Jarrassé.

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
