## [Decision Letter · Decision Letter 0]

1 Sep 2022

PONE-D-22-21171Kinematic analysis of impairments and compensatory motor behavior during prosthetic grasping in below-elbow amputees.PLOS ONE

Dear Dr. Roby-Brami,

Thank you for submitting your manuscript to PLOS ONE. After careful consideration, we feel that it has merit but does not fully meet PLOS ONE’s publication criteria as it currently stands. Therefore, we invite you to submit a revised version of the manuscript that addresses the points raised during the review process.

The Reviewers have asked for a number of changes. The requested changes seem reasonable to me, and I believe that you can prepare a suitable revision. I encourage you to do so.

We look forward to receiving your revised manuscript.

Kind regards,

Thomas A Stoffregen, PhD

Academic Editor

PLOS ONE

Journal Requirements:

"The authors thank all the participants to this study. Funding was supported by the Labex SMART (supported by French state funds managed by the ANR within the “Investissements d’Avenir” program under reference ANR-11-IDEX-0004-02). Adrienne Gouzien received an internship grant from the Institut Universitaire Ingéniérie de la Santé in Sorbonne University. "

"ARB and NJ received a funding by the Labex SMART (supported by French state funds managed by the ANR within the “Investissements d’Avenir” program under reference ANR-11-IDEX-0004-02). http://www.smart-labex.fr/

AG received an internship grant from the Institut Universitaire Ingéniérie de la Santé in Sorbonne University. https://iuis.sorbonne-universite.fr/

Reviewers' comments:

Reviewer's Responses to Questions

**Comments to the Author**

1. Is the manuscript technically sound, and do the data support the conclusions?

Reviewer #1: Partly

Reviewer #2: Yes

2. Has the statistical analysis been performed appropriately and rigorously? 

Reviewer #1: I Don't Know

Reviewer #2: Yes

3. Have the authors made all data underlying the findings in their manuscript fully available?

Reviewer #1: Yes

Reviewer #2: Yes

4. Is the manuscript presented in an intelligible fashion and written in standard English?

Reviewer #1: Yes

Reviewer #2: Yes

5. Review Comments to the Author

Reviewer #1: Review of PONE-D-22-21171

Kinematic analysis of impairments and compensatory motor behavior during prosthetic

grasping in below-elbow amputees

This manuscript describes a single experiment investigating the kinematics of reaching and grasping three targets at two distances among below elbow amputees using a myoelectric prosthetic.

The topic is interesting and important. However, I have a number of concerns about the methodology and data analysis, some of which prevent me from being able to assess the interpretation of the results. These concerns would need to be addressed in a revision.

My largest concern about the methodology and the results is that the data analyses themselves were unclear to me on a first reading. The section called ‘data analysis’ is really about the dependent variables more than it is about data analysis. The section called ‘statistics’ is somewhat opaquely about the choice of analysis. This made it difficult (for me anyway) to follow the description of the findings in the method section. I think that the authors need to be more straightforward about what is being analyzed, how it is being analyzed, why it is being analyzed that way, and what the results of each analysis show.

Minor concerns

1. Misunderstanding/misuse of the term ‘affordance’: Affordances are opportunities for behavior given the fit between animal and environment. Given that, I don’t know what it could mean that some objects ‘afford more…or less…comfortable affordances (p. 8, lines 15-155), or to ‘favor the handle affordance’ (p. 8, line 166), or that ‘…the different object affordances are taken over by the versatility of the fingers (p. 23, lines 533—534).

2. In addition, the authors refer to this study as an ‘observational case study’? Is this because of variation among participants in their ages, specifics of their amputation, prosthetic device, length of time using such devices, among other factors? Presumably participants used their own personal prosthetic devices (though this is never stated). If so, were there marked differences in these devices across participants?

3. The authors claim that “Current prosthetic medical devices do not provide direct sensory information about the subject's environmental context or the position of the prosthetic limb when manipulating an object.” It is unclear to me what ‘direct sensory information’ is. Are you referring to fact that the device itself is non-innervated? If so, this is not prerequisite for successful behavior. People use objects attached to the body as perceptual tools quite regularly. And they can even perceive properties of object wielded by neuropathic limbs

Silva, P. L., Harrison, S., Kinsella-Shaw, J., Turvey, M. T., & Carello, C. (2009). Lessons for dynamic touch from a case of stroke-induced motor impairment. Ecological Psychology, 21(4), 291–307. https://doi.org/10.1080/10407410903320926

Carello, C., Kinsella-Shaw, J., Amazeen, E. L., & Turvey, M. T. (2006). Peripheral neuropathy and object length perception by effortful (dynamic) touch: a case study. Neuroscience letters, 405(3), 159–163. https://doi.org/10.1016/j.neulet.2006.06.047

Carello, C., Silva, P. L., Kinsella-Shaw, J. M., & Turvey, M. T. (2008). Muscle-based perception: theory, research and implications for rehabilitation. Brazilian Journal of Physical Therapy, 12, 339-350.

4. When the authors claim that the control of myoelectric prosthetics “… remains particularly non physiological (sequential and delayed) and complex both to learn and to use” I recommend that they cite

van Dijk, L., van der Sluis, C. K., van Dijk, H. W., & Bongers, R. M. (2016a). Learning an EMG Controlled Game: Task-Specific Adaptations and Transfer. PloS one, 11(8), e0160817. https://doi.org/10.1371/journal.pone.0160817

van Dijk, L., van der Sluis, C. K., van Dijk, H. W., & Bongers, R. M. (2016b). Task-Oriented Gaming for Transfer to Prosthesis Use. IEEE transactions on neural systems and rehabilitation engineering : a publication of the IEEE Engineering in Medicine and Biology Society, 24(12), 1384–1394. https://doi.org/10.1109/TNSRE.2015.2502424

5. On page 18 (lines 405-407) the authors claim that “The examination of the kinematic details by comparison to well-known observations in healthy participants and scarce observations in amputees may provide clues for the analysis of these mechanisms.” So what are these well-known observations? And what are these clues?

6. Relatedly, the set of participants varied widely in experience using a prosthetic (from 3 months to 324 months). Were there any differences across the most and least experienced prosthetic users?

7. On page 18 (lines 412-413), the authors claim that “Physiological goal directed upper-limb movements are characterized by well-known invariants.” Invariants in what?

Page 6, lines 106-109. Awkward or confusing sentence(s)

Page 8, lines 153-155. Awkward or confusing sentence(s)

Page 8, lines 157-159. Awkward or confusing sentence(s)

Page 19, lines 446-448, Awkward or confusing sentence(s)

Page 20, lines 451, Awkward or confusing sentence(s)

Reviewer #2: The authors conducted a descriptive study measuring upper extremity trajectories during a reaching and grasping task using prosthetic and intact arms. Past assessment of mobility of prosthetic appendages was done mostly through clinical tests that provide information about performance in everyday tasks, but the kinematics of the movements are rarely recorded to screen for abnormal movement patterns and signs of inappropriate rehabilitation and usage of prostheses. The current study discovered significant differences in movement patterns (e.g. timing, range and variability of motion) between prosthetic and intact limbs. Amputees exhibited compensatory movement strategies that were not optimal. The study was described adequately, the analyses were appropriate, and the writing is mostly clear and concise.

Detailed comments:

Page 4, lines 55-56: Explain what you mean by “life project”

Page 10, lines 218-224: Please label in Figure 3 where t0, tv and tg are located in time in the exemplary trial presented in the figure.

Page 10, line 225: What feature of the movement is operationalized by the variable “the number of velocity peaks”? Complexity of movement? Variability? Please clarify.

Page 12, line 264 (and elsewhere): please calculate effect sizes for all reported statistical tests.

Page 13, lines 280-282, and Figure 4: please explain the huge range for the prosthetic arm for the far distance. Anything in particular causing this large range? Outliers? Is that why the effect of distance is non-significant on number of peaks?

Page 13, lines 299-300: “varied with the condition” is vague: which condition? Distance or arm, or both?

Page 17, line I don’t understand what “percent of variation” means. For example does 50% mean half as long as the non-amputated side? Please clarify how this variable is defined.

Page 24, lines 550-553: Was the weight of the prosthesis comparable to the weight of the missing limb portion? Can this be one of the reasons for the asymmetry between amputated and intact limb kinematics?

6. PLOS authors have the option to publish the peer review history of their article (what does this mean?). If published, this will include your full peer review and any attached files.

Reviewer #1: No

Reviewer #2: **Yes: **Alen Hajnal

---

## [Author Response · Author response to Decision Letter 0]

14 Oct 2022

Reviewer #1: Review of PONE-D-22-21171

My largest concern about the methodology and the results is that the data analyses themselves were unclear to me on a first reading. The section called ‘data analysis’ is really about the dependent variables more than it is about data analysis. The section called ‘statistics’ is somewhat opaquely about the choice of analysis. This made it difficult (for me anyway) to follow the description of the findings in the method section. I think that the authors need to be more straightforward about what is being analyzed, how it is being analyzed, why it is being analyzed that way, and what the results of each analysis show.

We tried to clarify the presentation of data analysis by adding a section “dependent variables” and by a more precise description of the statistics. 

“The analysis was performed according to the following steps. 

First, we analyzed the effect of Side and Distance on the values obtained by averaging the dependent variables over the three objects. Friedman analysis was used to test the effect of condition (4 conditions: Side x Distance). When Friedman analysis showed significant variations, Wilcoxon test was used for paired comparisons: separately the effect of Side (for both distances) and the effect of Distance (for both sides). 

Secondly: the effects of object were investigated by Friedman performed for each of the four combined side-distance condition and completed by paired comparisons between objects (Wilcoxon test). 

Effect sizes of paired comparisons were measured by the Cohen’s d (difference between means divided by the standard error of the difference).

Correlation analysis was performed to test the relationship between clinical data (age, delay from amputation, duration of prosthetic use, embodiment score) and kinematic variables. Kinematic variables were expressed as a percent of variation of the prosthetic side (P) by reference to the non-amputated side (NA) 

percent of variation=(P-NA)÷NA*100”

Minor concerns

1. Misunderstanding/misuse of the term ‘affordance’: Affordances are opportunities for behavior given the fit between animal and environment. Given that, I don’t know what it could mean that some objects ‘afford more…or less…comfortable affordances (p. 8, lines 15-155), or to ‘favor the handle affordance’ (p. 8, line 166), or that ‘…the different object affordances are taken over by the versatility of the fingers (p. 23, lines 533—534).

We agree that affordances are opportunities for behavior given the fit between animal and environment. Here we question the fit between the shape of the hand and that of the object to grasp. Since the prosthesis has different properties than the anatomical hand (shape, degrees of freedom, softness, friction.. ), we expect different grasping opportunities with the anatomical versus prosthetic hand according to the shape of the object. According to experimented occupational therapists, the easiest object to grasp by amputees is the kind of cardboard cone routinely used in rehabilitation. In contrast, amputees have difficulties to grasp a mug by the handle, the cylinder being intermediate. We explain this more clearly in the methods. In addition, we give some descriptive details on the kind of prostheses that were used by the amputees in this study. 

We also used the word affordance to describe some features of graspable objects as proposed by Arbib et al (e.g. in Oztop E, Arbib MA. Biol Cybern. 2002 Aug;87(2):116-40). In order to avoid conceptual ambiguity, we removed the reference to “affordance” for the description of object features. 

2. In addition, the authors refer to this study as an ‘observational case study’? Is this because of variation among participants in their ages, specifics of their amputation, prosthetic device, length of time using such devices, among other factors? Presumably participants used their own personal prosthetic devices (though this is never stated). If so, were there marked differences in these devices across participants?

We used the terminology “observational case study” by reference to the classification of medical studies. This means that there is no medical intervention nor population analysis.

Indeed, the patients have quite different personal characteristics. We apologize because the Table showing those important characteristics had been omitted during the submission procedure. It is now included in the revised manuscript. The patients used their own prosthesis, this is now mentioned in the text. There was no marked difference between the prostheses used by the participants: all the prostheses were tridigital with flexion-extension. Some prostheses included motorized wrist rotation (prono-supination), this now is indicated in the Table1 with clinical data. 

3. The authors claim that “Current prosthetic medical devices do not provide direct sensory information about the subject's environmental context or the position of the prosthetic limb when manipulating an object.” It is unclear to me what ‘direct sensory information’ is. Are you referring to fact that the device itself is non-innervated? If so, this is not prerequisite for successful behavior. People use objects attached to the body as perceptual tools quite regularly. And they can even perceive properties of object wielded by neuropathic limbs. 

Silva, P. L., Harrison, S., Kinsella-Shaw, J., Turvey, M. T., & Carello, C. (2009). Lessons for dynamic touch from a case of stroke-induced motor impairment. Ecological Psychology, 21(4), 291–307. https://doi.org/10.1080/10407410903320926

Carello, C., Kinsella-Shaw, J., Amazeen, E. L., & Turvey, M. T. (2006). Peripheral neuropathy and object length perception by effortful (dynamic) touch: a case study. Neuroscience letters, 405(3), 159–163. https://doi.org/10.1016/j.neulet.2006.06.047

Carello, C., Silva, P. L., Kinsella-Shaw, J. M., & Turvey, M. T. (2008). Muscle-based perception: theory, research and implications for rehabilitation. Brazilian Journal of Physical Therapy, 12, 339-350.

We agree with the reviewer that we were unclear. We indeed meant that the device is not innervated and that the lack of direct sensory information contributes to the difficulties of prosthetic use, particularly during the phase of grasping. There is a difference between myoelectric and body-powered prostheses where hand closure is ensured by pulling a harness strapped around the shoulder. In this later case, object contact and the force of grasping can be directly felt through the harness. This direct perception of grasping is not possible when the finger flexion is motorized. We are aware that hand-held objects can be perceptual tools and we thank the reviewer to have raised this point. We have clarified and completed the topic of sensory information in the introduction and added the reference to dynamic touch. However, did not enter into the question of perceptual tool in order to remain focused on the topic. 

4. When the authors claim that the control of myoelectric prosthetics “… remains particularly non physiological (sequential and delayed) and complex both to learn and to use” I recommend that they cite

van Dijk, L., van der Sluis, C. K., van Dijk, H. W., & Bongers, R. M. (2016a). Learning an EMG Controlled Game: Task-Specific Adaptations and Transfer. PloS one, 11(8), e0160817. https://doi.org/10.1371/journal.pone.0160817

van Dijk, L., van der Sluis, C. K., van Dijk, H. W., & Bongers, R. M. (2016b). Task-Oriented Gaming for Transfer to Prosthesis Use. IEEE transactions on neural systems and rehabilitation engineering : a publication of the IEEE Engineering in Medicine and Biology Society, 24(12), 1384–1394. https://doi.org/10.1109/TNSRE.2015.2502424

We thank the reviewer for his/her suggestion of references that were overlooked. They are now indicated in the text. We also added two additional references (Sobuh et al, 2014 and Cipriani et al, 2008) about the visual and cognitive control of a myoelectric prosthesis.

5. On page 18 (lines 405-407) the authors claim that “The examination of the kinematic details by comparison to well-known observations in healthy participants and scarce observations in amputees may provide clues for the analysis of these mechanisms.” So what are these well-known observations? And what are these clues?

We agree that this sentence is not informative and we replaced it by “The extensive literature on the kinematics of reaching to grasp in healthy subjects contrasts with scarce observations in amputees using a prosthesis. The observations in healthy subjects and amputated patients are described below in two sections “Temporal organization” and “Spatial organization of hand trajectory and inter-joint configuration for grasping”. 

6. Relatedly, the set of participants varied widely in experience using a prosthetic (from 3 months to 324 months). Were there any differences across the most and least experienced prosthetic users?

We agree that the small number of participants is a limitation of our study. As explained in the result section, paragraph individual factors “We explored potential relationships between motor strategies (trunk flexion for reaching, particularities of hand orientation) and clinical data, in particular the proximo/distal level of the amputation but we did not find any pertinent relation with clinical variables. The limited number of participants did not allow to make further statistics”. We add the precision that we did not find any correlation between kinematic variables and clinical data. 

7. On page 18 (lines 412-413), the authors claim that “Physiological goal directed upper-limb movements are characterized by well-known invariants.” Invariants in what?

We agree that the word is misleading and questionable, it is replaced by “characteristics”

Page 6, lines 106-109. Awkward or confusing sentence(s)

The sentence is now replaced by: “From a clinical point of view, it is important to consider compensatory strategies and to differentiate between useful/unavoidable and harmful/avoidable ones. Indeed, certain compensations are inevitable when using prostheses. A better understanding of their mechanisms would help to manage compensatory strategies during rehabilitation for the prevention and treatment of musculoskeletal disorders”.

Page 8, lines 153-155. Awkward or confusing sentence(s)

The sentence is now replaced by “These objects were chosen after discussion with occupational therapists examining the grasping affordances offered by various objects to anatomical or tradigital prosthetic hands. The easiest object to grasp by amputees is a cone, routinely used in rehabilitation. In contrast, amputees have difficulties to grasp a mug by the handle, the cylinder being intermediate”.

Page 8, lines 157-159. Awkward or confusing sentence(s)

The sentence is now replaced by “MRD was measured before the session when the participant was comfortably sitting, between the patient belly and the most distant forward distance he/she is able to reach with the center of the prosthetic hand.” 

Page 19, lines 446-448, Awkward or confusing sentence(s) 

The sentence is now replaced by “Moreover, the loss of mobility at wrist and finger levels probably participates to the impairment of grasping“.

Page 20, lines 451, Awkward or confusing sentence(s)

The sentence is now replaced by “There are few kinematic studies of prosthetic reaching and grasping and most include a very small number of heterogeneous participants.”

Reviewer #2: (Alen Hajnal)

The authors conducted a descriptive study measuring upper extremity trajectories during a reaching and grasping task using prosthetic and intact arms. Past assessment of mobility of prosthetic appendages was done mostly through clinical tests that provide information about performance in everyday tasks, but the kinematics of the movements are rarely recorded to screen for abnormal movement patterns and signs of inappropriate rehabilitation and usage of prostheses. The current study discovered significant differences in movement patterns (e.g. timing, range and variability of motion) between prosthetic and intact limbs. Amputees exhibited compensatory movement strategies that were not optimal. The study was described adequately, the analyses were appropriate, and the writing is mostly clear and concise.

We thank the reviewer for his positive appreciation. 

Detailed comments:

Page 4, lines 55-56: Explain what you mean by “life project”. 

We tried to be more precise. “The patient's living conditions, leisure time and professional activity are considered to define the rehabilitation and fitting project”.

Page 10, lines 218-224: Please label in Figure 3 where t0, tv and tg are located in time in the exemplary trial presented in the figure.

We thank you for the suggestion, the labels have been added. 

Page 10, line 225: What feature of the movement is operationalized by the variable “the number of velocity peaks”? Complexity of movement? Variability? Please clarify. 

The number of velocity peaks is a way to measure the smoothness of the movement. Smooth reaching movements in healthy subjects are characterized by one only bell-shaped velocity peak. These precisions are now added. 

Page 12, line 264 (and elsewhere): please calculate effect sizes for all reported statistical tests.

We calculated the effect size for all significant two-by-two comparisons using Cohen’s d. This is now indicated in the Method and Result sections. 

Page 13, lines 280-282, and Figure 4: please explain the huge range for the prosthetic arm for the far distance. Anything in particular causing this large range? Outliers? Is that why the effect of distance is non-significant on number of peaks?

The large dispersion was due to outliers. Two patients had a particularly “shaky” behavior that we cannot particularly explain. This fact probably explains the lack of significance. 

Page 13, lines 299-300: “varied with the condition” is vague: which condition? Distance or arm, or both?

The variations were considered according to both side and distance. This is now indicated. 

Page 17, line I don’t understand what “percent of variation” means. For example does 50% mean half as long as the non-amputated side? Please clarify how this variable is defined.

We have clarified this variable in the Methods section. “Correlation analysis was performed to test the relationship between clinical data (age, delay from amputation, duration of prosthetic use, embodiment score,) and kinematic variables. Kinematic variables were expressed as a percent of variation of the prosthetic side (P) by reference to the non-amputated side (NA) 

percent of variation=(P-NA)÷NA*100"

Page 24, lines 550-553: Was the weight of the prosthesis comparable to the weight of the missing limb portion? Can this be one of the reasons for the asymmetry between amputated and intact limb kinematics?

We agree that the weight was asymmetric between amputated and intact limb. This is now explicit “Indeed, the weight distribution of the prosthesis differs from that of the missing limb with a shift in weight towards the most distal part of the prosthesis. This is due to the prosthetic hand and the motor which are the heaviest parts of the prosthesis and are the most distal. This difference may contribute to the asymmetry observed between amputated and intact limb kinematics”.

---

## [Decision Letter · Decision Letter 1]

6 Nov 2022

Kinematic analysis of impairments and compensatory motor behavior during prosthetic grasping in below-elbow amputees.

PONE-D-22-21171R1

Dear Dr. Roby-Brami,

We’re pleased to inform you that your manuscript has been judged scientifically suitable for publication and will be formally accepted for publication once it meets all outstanding technical requirements.

Kind regards,

Thomas A Stoffregen, PhD

Academic Editor

PLOS ONE

Additional Editor Comments (optional):

Reviewers' comments:

Reviewer's Responses to Questions

**Comments to the Author**

1. If the authors have adequately addressed your comments raised in a previous round of review and you feel that this manuscript is now acceptable for publication, you may indicate that here to bypass the “Comments to the Author” section, enter your conflict of interest statement in the “Confidential to Editor” section, and submit your "Accept" recommendation.

Reviewer #1: All comments have been addressed

Reviewer #2: All comments have been addressed

2. Is the manuscript technically sound, and do the data support the conclusions?

Reviewer #1: Yes

Reviewer #2: Yes

3. Has the statistical analysis been performed appropriately and rigorously? 

Reviewer #1: Yes

Reviewer #2: Yes

4. Have the authors made all data underlying the findings in their manuscript fully available?

Reviewer #1: Yes

Reviewer #2: Yes

5. Is the manuscript presented in an intelligible fashion and written in standard English?

Reviewer #1: Yes

Reviewer #2: Yes

6. Review Comments to the Author

Reviewer #1: The authors have successfully addressed all of my concerns. I appreciated their efforts to do so. I can now recommend that the manuscript be accepted for publication

Reviewer #2: The authors addressed all of my concerns. I think this research adds to the literature on amputees motor rehabilitation.

7. PLOS authors have the option to publish the peer review history of their article (what does this mean?). If published, this will include your full peer review and any attached files.

Reviewer #1: No

Reviewer #2: **Yes: **Alen Hajnal

---

## [Editor Report · Acceptance letter]

10 Nov 2022

PONE-D-22-21171R1 

Kinematic analysis of impairments and compensatory motor behavior during prosthetic grasping in below-elbow amputees. 

Dear Dr. Roby-Brami:

I'm pleased to inform you that your manuscript has been deemed suitable for publication in PLOS ONE. Congratulations! Your manuscript is now with our production department. 

Kind regards, 

on behalf of

Dr. Thomas A Stoffregen 

Academic Editor

PLOS ONE